# ZerO Initialization: Initializing Residual Networks with only Zeros and Ones

## Abstract

Deep neural networks are usually initialized with random weights, with adequately selected initial variance to ensure stable signal propagation during training. However, there is no consensus on how to select the variance, and this becomes challenging especially as the number of layers grows. In this work, we replace the widely used random weight initialization with a *fully deterministic* initialization scheme ZerO, which initializes residual networks with only *zeros and ones*. By augmenting the standard ResNet architectures with a few extra skip connections and Hadamard transforms, ZerO allows us to start the training from zeros and ones entirely. This has many benefits such as improving reproducibility (by reducing the variance over different experimental runs) and allowing network training without batch normalization. Surprisingly, we find that ZerO achieves state-of-the-art performance over various image classification datasets, including ImageNet, which suggests random weights may be unnecessary for modern network initialization [1].

## 1 Introduction

An important question in training deep neural networks is how to initialize the weights. When Rumelhart et al. first introduced backward propagation, they initialized the weights randomly to break symmetries among different parameters and activations. Currently, random weight initialization is the de-facto practice across all architectures and tasks.

However, choosing the *variance* of the initial distribution or equivalently the *scale* of the initial weights is a delicate balance when training deep neural networks. Too large a scale can lead to excessive amplification of the activations propagating through the network, resulting in *exploding gradients* during training. On the other hand, if the weights are initialized too small the activations may not propagate at all, resulting in *vanishing gradients*, which is especially an issue as the depth of the network grows.

He et al. (a); Glorot and Bengio study the propagation of variance in the forward and backward passes at initialization, using it as a proxy for the magnitudes of the gradients. By choosing scale parameters that depend on the width of the neural networks, to upper bound the variance, they design initialization schemes that avoid exploding gradients. However, this still does not fully address the issue of vanishing gradients because it is not easy to estimate the lower bounds on the variance.

In order to design networks robust to vanishing gradients at initialization, He et al. (b) introduced *skip connections* that take the activations of a layer, and add them to the next layer. Given enough skip connections, the activations can, in principle, always be propagated through the entire network. Thus, skip connections allow us to mitigate the vanishing gradient problem and possibly avoid it completely.

Zhang et al. (a); Yang and Schoenholz observe that initializing the residual branches (i.e., the branch that is *not* the skip connection) at (or close to) zero benefits signal propagation and optimization. To achieve this, Zhang et al. (a) initialize the last layer of each residual branch to zero, which is

---

[1]We share our code via an anonymous link here.

also shown to stabilize training without batch normalization. However, previous studies rely on random weights to propagate signals to layers without skip connections, which still requires the consideration of the weight variance at initialization.

**Our approach:** In this work, we show that **all randomness in the weight initialization can be removed,** resulting in a fully deterministic initialization scheme ZerO, which initializes residual networks with only *zeros and ones*. As illustrated in Figure 1, ZerO achieves it by adding extra skip connections to enable non-zero signal propagation throughout the network. Additionally, it applies the Hadamard transform to avoid the training degeneracies in the latent space when there is a width expansion across layers. Moreover, our approach does not add any new learnable parameters to the standard architectures and does not involve any significant computational overhead. We find that ZerO achieves state-of-the-art performance over various image classification tasks.

**Our contributions are summarized as follows:**

1. We carefully identify the training difficulties that occur when initializing residual networks to zero, including 1) the dead neuron problem—addressed by employing skip connections at each layer, and 2) the degeneracies due to width expansion—addressed by applying the Hadamard transform.

2. We propose ZerO initialization for ResNet as an example to show how skip connections enable a deterministic initialization. Our experiments show that ResNet with ZerO achieves state-of-the-art results over various image classification tasks

3. We find ZerO works well without batch normalization during training. With proper regularization, ZerO beats random initialization methods such as Kaiming initialization while almost matching the batch normalization baseline.

4. We observe that compared to random initialization methods, the deterministic ZerO initialization achieves 20%-40% lower standard deviation over repeated experiments, and thus ZerO enjoys the advantage of high reproducibility.

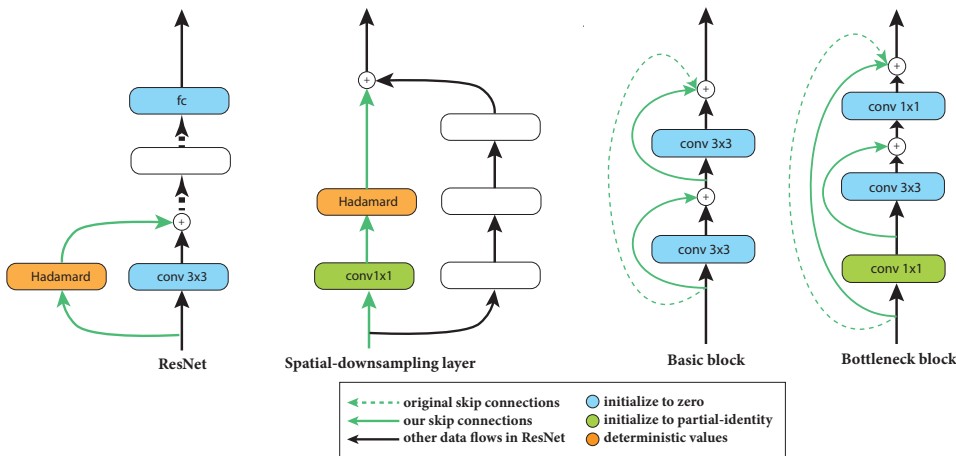

Figure 1: *ZerO initialization for ResNet.* We add extra skip connections and modify their locations in the standard ResNet. We also apply Hadamard transforms (defined at Definition 1) when there is an expansion in the channel dimension. We omit Relu and batch normalization here, see Figure 4 for a detailed design. A partial-identity 1x1 convolutional kernel is defined at Equation 1.

## 2   WHEN DOES A ZERO-INITIALIZED NEURAL NETWORK WORK?

In this section, we discuss when a zero-initialized neural network works by identifying the training difficulties that occur when initializing the weights to zero. We also propose adequate methodologies addressing these problems in order to achieve zero initialization without damaging the performance.

We use the fully connected neural network as a proxy to illustrate the difficulties and our idea. Consider a problem of learning a function $\hat{y} = \mathcal{F}(x)$ with parameters $W$ by minimizing the squared error $\mathcal{L} = \frac{1}{2}\|y - \hat{y}\|_2^2$. $(x, y) \in \mathbb{R}^{N_x \times N_y}$ represent the sample data, and the learning

is accomplished via gradient descent. $\mathcal{F}(\boldsymbol{x})$ is a fully connected neural network with $L$ layers: $\mathcal{F}(\boldsymbol{x}) = \varphi(\boldsymbol{W}^L) \circ ... \circ \varphi(\boldsymbol{W}^1) \circ \boldsymbol{x}$, where $\circ$ is composition operator, $\boldsymbol{W}^l \in \mathbb{R}^{P^l \times Q^l}$ for $l \in 1, ..., L$, and $\varphi$ is an nonlinearity. We ignore bias terms, and denote $\boldsymbol{x}^l = \varphi(W^l \boldsymbol{x}^{l-1})$ as the activations of the $l$th layer. We also define $\boldsymbol{W}_{i,:}^l$ and $\boldsymbol{W}_{:,j}^l$ to be $i$th row and $j$th column of the matrix $\boldsymbol{W}^l$.

## 2.1 Skip connections solve dead neuron problem

For simplicity, we first assume that $\mathcal{F}(\boldsymbol{x})$ is a linear network with a fixed dimension, such that $N_x = N_y = P^l = Q^l$ for any $l$, and $\varphi$ is an identity mapping. In this case, if we initialize the weights to zero, every layer will output zero constantly no matter what the inputs are. On the other hand, every $\boldsymbol{W}^l$ receives zero derivatives in the backward pass, and thus there is no weight that can move out of the zero during the entire training. This extreme case of the vanishing gradient problem, well-known under the name *dead neuron problem*, is caused by zero-initialized $\mathcal{F}(\boldsymbol{x})$ only generating zero signals at initialization.

Skip connections help propagate signals through the network and thus mitigate the vanishing gradient problem. A natural approach towards practical zero initialization is therefore to equip *every* layers of the network with skip connections. This ensures that the signal is propagated even with zero weights, with each layer applying a simple identity function to the activations. For example, equipping each layer of $\mathcal{F}(\boldsymbol{x})$ with a skip connection, we obtain

$$\mathcal{F}(\boldsymbol{x}) = (\boldsymbol{W}^L + \boldsymbol{I})...(\boldsymbol{W}^1 + \boldsymbol{I})\boldsymbol{x}.$$

The derivatives with respect to weight matrices are then given as

$$\frac{\partial \mathcal{L}}{\partial \boldsymbol{W}^1} = ((\boldsymbol{W}^L + \boldsymbol{I})...(\boldsymbol{W}^2 + \boldsymbol{I}))^\top (y - \hat{y}) \, \boldsymbol{x}^\top,$$

$$\frac{\partial \mathcal{L}}{\partial \boldsymbol{W}^l} = ((\boldsymbol{W}^L + \boldsymbol{I})...(\boldsymbol{W}^{l+1} + \boldsymbol{I}))^\top (y - \hat{y}) \, ((\boldsymbol{W}^{l-1} + \boldsymbol{I})...(\boldsymbol{W}^1 + \boldsymbol{I}) \, \boldsymbol{x})^\top \quad \text{for } l \in 2, ..., L-1,$$

$$\frac{\partial \mathcal{L}}{\partial \boldsymbol{W}^L} = (y - \hat{y}) \, ((\boldsymbol{W}^{L-1} + \boldsymbol{I})...(\boldsymbol{W}^1 + \boldsymbol{I}) \, \boldsymbol{x})^\top.$$

As zero-initialized weights initialize $\mathcal{F}(\boldsymbol{x})$ as an identity mapping, the loss $\mathcal{L}$ at initialization is the error when directly using the inputs as the predicted outputs $\hat{y} = \boldsymbol{x}$. Therefore, the derivatives at initialization uniformly become $\frac{\partial \mathcal{L}}{\partial \boldsymbol{W}^l} = (\boldsymbol{y} - \boldsymbol{x}) \, \boldsymbol{x}^\top$ for every $\boldsymbol{W}^l$. This is because the skip connections are applied to every layer, and thus the error derivatives flow back to every matrix equivalently. With the help of the skip connections, each weight matrix $\boldsymbol{W}^l$ receives non-zero derivatives, which break the dead neuron problem for the entire network.

This also works for residual networks with nonlinearity, as long as the derivative of the nonlinearity exists at zero. Although there exists nonlinearity that is non-differentiable at zero such as Relu (Xu et al.), this can be solved easily by utilizing its subderivatives at zero. For example, we define the derivative of Relu to be its subderivative one at zero, which allows the signals to be identically propagated through the Relu. We will apply this special modification to Relu in our later experiments.

We note that in our example above, at the first iteration, all weight matrices receive the same derivatives. However, as training progresses, the gradient that a weight matrix receives depends on its position in the network, and thus the full capacity of the network is utilized.

## 2.2 Width expansion leads to training degeneracy

For the deep residual networks discussed above, the dimensions of all hidden layers are equal to the input and output dimensions. However, for modern residual networks, the hidden dimensions are usually varying and larger than the input dimension, such as the large channel dimension in intermediate layers of ResNet. We find that for these networks, directly initializing weights to zero leads to another kind of dead neuron problem, which can not be avoided even with skip connections. To distinguish it from the previous dead neuron problem, we will refer to this problem as *training degeneracy*.

To simplify the presentation, we explain this phenomenon in the setting where the nonlinearities are applied to the outputs of skip connections from previous layers. However, an equivalent phenomenon is present in the case where skip connections skip the nonlinearity as well, which is the variant we are using in Section 3.

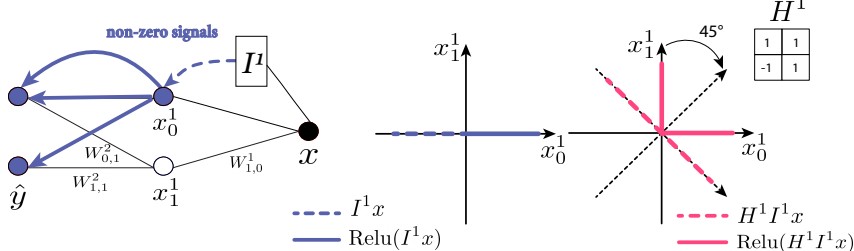

Figure 2: **Left:** the forward dynamic of the zero-initialized example network at initialization. Solid and dashed blue lines represent non-zero signals. **Right:** we represent the signals in the first layer at initialization on a standard 2-dimensional basis. Dashed lines represent the signals before the Relu and solid lines represent the signals after the Relu, which are activations $x^1$.

To understand the cause of this degeneracy, we consider a simple 2-layer residual network: $\mathcal{F}(\boldsymbol{x}) = \text{Relu}(\boldsymbol{W}^2 + \boldsymbol{I}^2) \circ \text{Relu}(\boldsymbol{W}^1 + \boldsymbol{I}^1)\boldsymbol{x}$ where $\boldsymbol{W}^1, \boldsymbol{I}^1 \in \mathbb{R}^{2 \times 1}$ and $\boldsymbol{W}^2, \boldsymbol{I}^2 \in \mathbb{R}^{2 \times 2}$. $N_x$ and $N_y$ are changed to 1 and 2 accordingly, and we use Relu as the activation. Because the dimension is increased in the network, we apply zero padding to the skip connection $\boldsymbol{I}^1$ to match the dimension, where $\boldsymbol{I}^1_{0,0} = 1$ and $\boldsymbol{I}^1_{1,0} = 0$. This duplicates the previous activations while padding additional dimensions using zero, which is a standard parameter-free method applied in ResNet (He et al., c).

For such a network, there exists a training degeneracy when directly initializing all the weights to zero. As illustrated in Figure 2, at initialization, the second dimension of the first layer activations $x^1_1$ receives a zero signal padded by $\boldsymbol{I}^1$, and it propagates the zero signal to $\hat{y}$ as well. During backpropagation, $\frac{\partial \mathcal{L}}{\partial \boldsymbol{W}^2_{1,1}}$ and $\frac{\partial \mathcal{L}}{\partial \boldsymbol{W}^2_{0,1}}$ are zero because $x^1_1 = 0$. Also, $\frac{\partial \mathcal{L}}{\partial \boldsymbol{W}^1_{1,0}}$ is zero as it receives the zero derivative from $\frac{\partial \mathcal{L}}{\partial x^1_1}$. Because $x^1_1$ and $\frac{\partial \mathcal{L}}{\partial x^1_1}$ keep receiving the zero signal, $\boldsymbol{W}^1_{1,0}$, $\boldsymbol{W}^2_{0,1}$ and $\boldsymbol{W}^2_{1,1}$ are bound to be zero *for the entirety of training*, restricting the expressivity of the network.

### 2.3 HADAMARD TRANSFORM AVOIDS TRAINING DEGENERACY

The reason for the degeneracy is that the skip connection $\boldsymbol{I}^1$ with zero padding is only able to map the input $\boldsymbol{x}$ into *a low-dimensional subspace* of $\boldsymbol{x}^1$. As illustrated in Figure 2, when initializing all the weights to zero, $\boldsymbol{I}^1$ maps the single dimension $\boldsymbol{x}$ to a horizontal line (dashed line) in the two-dimensional space. Because this subspace is aligned with the standard basis, its linear dimension is invariant under application of the Relu, or any component-wise nonlinearity. As the activations $\boldsymbol{x}^1$ stay in a 1-dimensional subspace span, its associated derivatives stay in such a subspace as well. This eventually causes the zero-initialized weights $W^1$ to be updated into a low-dimensional space, as they solely depend on the low-dimensional derivatives.

To ensure that the first hidden space receives high-dimensional activations $\boldsymbol{x}^1$, we propose to apply a *Hadamard transform* $\boldsymbol{H}^1$ after $\boldsymbol{I}^1$ to spread the information into a new space.

The Hadamard transform is an example of the generalized family of Fourier transforms that performs an orthogonal linear operation on $2^m$ real numbers (Pratt et al.). It consists of a Hadamard matrix and a normalization factor. A Hadamard matrix is defined as follows:

**Definition 1** (Hadamard matrix). For any Hadamard matrix $\boldsymbol{H}_m \in \mathbb{R}^{2^m \times 2^m}$ where $m$ is a positive integer, we define $\boldsymbol{H}_0 = 1$ by the identity, and the matrix with large $m$ is defined recursively:

$$\boldsymbol{H}_m = \begin{pmatrix} \boldsymbol{H}_{m-1} & \boldsymbol{H}_{m-1} \\ \boldsymbol{H}_{m-1} & -\boldsymbol{H}_{m-1} \end{pmatrix} = \begin{pmatrix} 1 & 1 & 1 & 1 & \dots \\ 1 & -1 & 1 & -1 & \dots \\ 1 & 1 & -1 & -1 & \dots \\ 1 & -1 & -1 & 1 & \dots \\ \vdots & \vdots & \vdots & \vdots & \ddots \end{pmatrix} \in \mathbb{R}^{2^m \times 2^m}.$$

We rescale the Hadamard matrix by $2^{-(m-1)/2}$, resulting in the orthonormal Hadamard transform. The Hadamard transform rotates the standard basis into a new basis, such that each element of the new basis is equally weakly aligned with every element of the standard basis. For example, in a two-dimensional space, the Hadamard transform rotates the standard basis by an angle of 45 degree, as illustrated in Figure 2 [2].

---

[2]We use a column-order-reversed Hadamard matrix in the example to simplify the presentation. This is equivalent to using a standard Hadamard matrix with order-reversed $\boldsymbol{I}^1$, without affecting our conclusions.

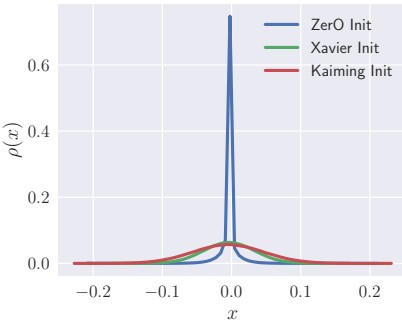 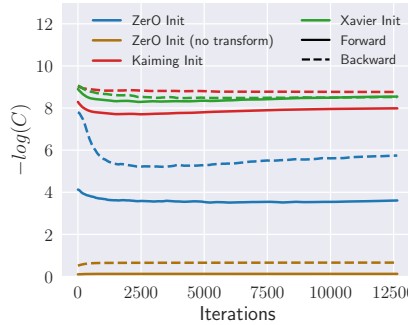

Figure 3: **Left:** final weight distributions. **Right:** weight correlations during training. Each setting achieves 98% test accuracy on MNIST dataset except for ZerO Init (no transform).

When applying component-wise nonlinearities to the spaces spanned by the elements of this basis, their linear dimension increases. In our example, after applying the Hadamard matrix $\boldsymbol{H}^1$ to $\boldsymbol{I}^1\boldsymbol{x}$, it forms a new space that is not aligned with the standard basis. Under the application of the Relu, the new space (i.e., $\boldsymbol{H}^1\boldsymbol{I}^1\boldsymbol{x}$) is spread into a 2-dimensional span in $\boldsymbol{x}^1$, which ensures that neither $\boldsymbol{x}_0^1$ nor $\boldsymbol{x}_1^1$ stay at zero forever. Therefore, with the Hadamard transform, the activation $\boldsymbol{x}^1$ can propagate the non-zero signals to the next layer. This ensures the derivatives of $\boldsymbol{W}_{1,1}^2$ and $\boldsymbol{W}_{0,1}^2$ to be non-zero, and thus breaks the training degeneracy.

## 2.4 Measuring the weight distributions and training degeneracy

To verify the phenomenon explained above empirically, we train a fully connected residual network on the MNIST dataset to check the existence of the degeneracy and measure the weight distribution (Lecun et al.). Our network has three layers with hidden dimensions larger than the input dimension. The details of our setting are introduced in the appendix.

We use empirical weight correlations as a proxy to verify the existence of the degeneracy, which is similar to the measurement conducted by Blumenfeld et al.. High weight correlations indicate that degeneracy may exist. For each weight matrix, we measure the correlations between row vectors and column vectors, respectively. For example, given a $M \times N$ weight matrix $\boldsymbol{W}$, we define forward correlations (between rows) $C_f$ and backward correlations (between columns) $C_b$ as follows:

$$C_f = \frac{1}{M(M-1)} \sum_i^M \sum_{i \neq j}^M \frac{\boldsymbol{W}_{i,:} \cdot \boldsymbol{W}_{j,:}}{\|\boldsymbol{W}_{i,:}\|_2 \|\boldsymbol{W}_{i,:}\|_2},$$

$$C_b = \frac{1}{N(N-1)} \sum_i^N \sum_{i \neq j}^N \frac{\boldsymbol{W}_{:,i} \cdot \boldsymbol{W}_{:,j}}{\|\boldsymbol{W}_{:,i}\|_2 \|\boldsymbol{W}_{:,j}\|_2},$$

where $A \cdot B$ denotes a dot product of vectors. We measure the weight correlations for the zero-initialized network with or without the Hadamard transform, and compare them with popular initialization methods proposed by Glorot and Bengio; He et al. (a). As shown in Figure 3, the Hadamard transform largely decouples the weight correlations compared to the network without the transform, which suggests that the Hadamard transform avoids the problem of training degeneracy.

In addition, we measure the final weight distributions over various initialization methods. As shown in Figure 3, the weight variance generated by our initialization is significantly lower than the variances generated by randomized methods. We believe that the fact that most of our weights are close to zero may help to train sparse neural networks with techniques such as weight pruning(Han et al.).

## 3 ZerO Initialization on Convolutional Residual Networks

In the last section, we found that training a zero-initialized fully connected neural network is possible when the network structure satisfies *two design principles*: 1) the skip connection is associated to every fully connected layer and 2) the Hadamard transform is applied to every increasing-dimension skip connection. We now propose *ZerO* by applying these design principles to the well-developed ResNet architectures.

It is a natural direction to apply our proposed design principles to modern convolutional networks such as ResNet, as they usually have skip connections between residual blocks to allow the signals to easily propagate through the network. However, modern convolutional networks require more consideration than simple plain networks because they have more complex designs and operations, such as dimension-varied residual block (e.g., bottleneck block), pooling operations, and batch normalization (He et al., c; Ioffe and Szegedy). To initialize ResNet deterministically, we propose ZerO that fully addresses the difficulties above by applying the following four steps.

> ### ZerO Initialization for ResNet
>
> 1. **Add additional residual connections.**
> 2. **Apply Hadamard transform to dimension-increasing residual connections.**
> 3. **Initialize residual blocks to zero and one.**
> 4. **Initialize both the first convolutional layer and last classification layer to zero.**

We now explain these steps in more detail.

We first focus on the design of the residual blocks, which is the core component of ResNet architectures. We adopt the pre-activation block design proposed by He et al. (b), where the identity mappings (without nonlinearity) are applied as the skip connections. We first consider modifying the basic block by using the skip connections to avoid the dead neuron problem. As the channel dimension is fixed within each basic block, we can directly apply identity mappings to every convolutional block, which contains a convolutional layer, Relu activation, and batch normalization. This is illustrated in Figure 1.

**Double-residual bottleneck block.**  The bottleneck block proposed by He et al. (c) is used to reduce the computational complexity when training deep models on large-scale datasets. The block consists of two 1x1 and one 3x3 convolutions where 1x1 convolutions are used to reduce and increase the input and output dimensions of the 3x3 convolution. Because the dimensions are varied within each bottleneck block, applying skip connection to every convolution leads to information loss. This is because the skip connection only can propagate a part of activations when applying it to a layer that reduces the channel dimension.

To address this problem, we only apply the skip connections between layers that share the same dimension. As illustrated in Figure 1, we keep the original skip connection between the first and the last 1x1 convolutions and only add another skip connection to the 3x3 one. This design can be viewed as building a residual layer inside another residual layer, which does not cause the information loss as we mentioned above. However, directly initializing all convolutions to zero is not applicable, as there is no skip connection between the first 1x1 convolution and the 3x3 convolution.

**Partial-identity initialization.**  To address this, we initialize each 1x1 convolution as a *partial identity* matrix, as it can be viewed as a linear projection over the channel dimension. For a $1 \times 1 \times c_{in} \times c_{out}$ convolutional kernel $K$, where $c_{in}$ is the number of input channels and $c_{out}$ is the number of output channels. Initializing $K$ is equivalent to initializing a $c_{in} \times c_{out}$ matrix $\hat{K}$. As $c_{in}$ and $c_{out}$ may be different, we initialize the largest square matrix in $\hat{K}$ as identity matrix and initialize the rest components to zero. This is illustrated in Figure 4. Specifically, let $c_{min}$ to be the minimum of $c_{in}$ and $c_{out}$, the matrix $\hat{K}$ is initialized as follows:

$$
\begin{aligned}
\hat{K}_{i,i} &= 1 \quad \text{for } 0 \le i \le c_{min} - 1, \\
\hat{K}_{i,j} &= 0 \quad \text{for } 0 \le i \le c_{in} - 1, 0 \le j \le c_{out} - 1, \text{ and } j \ne i.
\end{aligned}
\tag{1}
$$

When $c_{in} \le c_{out}$, this essentially initializes the linear projection as an identity mapping with zero padding. This allows signals to be propagated through the layer while initializing the weights deterministically. Therefore, by initializing the first 1x1 convolution as a partial-identity matrix, the rest convolutions can be initialized to zero without damaging the signal propagation.

**Hadamard transform for dimension-increasing skip connections.**  Because the channel dimension is increased gradually in ResNet, there are various types of dimension-increasing skip connec-

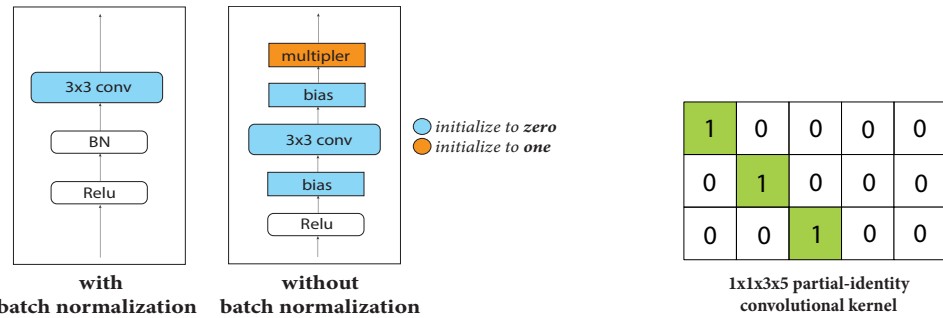

Figure 4: **Left:** associated operations for each convolutional layer. **Right:** an illustration of a 1x1x3x5 partial-identity convolutional kernel.

tions developed, including simple identity mapping with zero padding and linear projection using 1x1 convolution. Although directly initializing 1x1 convolution to zero leads to zero signals, we can follow our previous design to initialize it as a partial-identity matrix, which also forms it as an identity mapping with zero padding. However, as discussed in Section 2, there is a training degeneracy when applying the zero padding operation during the width expansion. Therefore, we apply Hadamard transform for every dimension-increasing skip connection to solve the degeneracy, as illustrated in Figure 1.

In addition, depending on the channel dimension, we may apply a dimension-increasing skip connection and a Hadamard transform for the first convolution of the ResNet. For the last fully connected layer, we can safely initialize it to zero without any modification, as it receives both non-zero inputs and derivatives at initialization. For batch normalization layers, we initialize the scale as one and the bias as zero. For other operations in ResNet, such as average or max pooling, we keep them by default as they do not require any learnable parameters.

We note that applying Hadamard transforms does not introduce a significant overhead as the transform is computationally efficient. As defined at Definition 1, a Hadamard matrix only consists of -1 and 1. Therefore, except for the scaling operation, it does not require any multiplication as sign flips are all it takes. It can be implemented efficiently with only $O(nlog(n))$ complexity (Fino and Algazi).

**Training without batch normalization.** As motivated in the introduction, ZerO may help train neural networks without batch normalization, because both ZerO and batch normalization bias the residual network as an identity mapping at initialization. Therefore, we propose removing the batch normalization using learnable scalar multipliers and biases, which follows the design in (De and Smith; Zhang et al., a). This is illustrated in Figure 4.

## 4 EXPERIMENTS

In this section, we empirically evaluate ZerO on CIFAR-10 and ImageNet datasets, and we also benchmark it under normalization-free settings. We evaluate ResNet-18 on CIFAR-10 and ResNet-50 on ImageNet (Krizhevsky; Deng et al.). Both ResNet structures follow the design from He et al. (c), which includes batch normalization by default.

**Hyperparameter settings.** We find that *ZerO can fully utilize the default hyperparameters*, which include a learning rate of $0.1$, a momentum of $0.9$, and a weight decay of $0.0001$. In addition, we observe the learning rate warmup is essential for ZerO to achieve a large maximal learning rate, as most of the weights start from the exact zero. We warm up the learning rate with 5 and 10 epochs for ImageNet and CIFAR-10, respectively.

**Fast convergence speed.** We observe that ZerO achieves a faster convergence speed than standard random initialization methods. As shown in Figure 5, at the initial training stage, ZerO converges faster than Xavier and Kaiming initialization when training ResNet-50 on ImageNet. This is because

ZerO forms ResNet as an identity mapping at initialization, which is an important property to achieve optimal signal propagation (Yang and Schoenholz).

We present our main results that compare architectures and initialization schemes on CIFAR-10 and ImageNet. For each dataset, all experiments use the same hyperparameter settings by default. Each experiment is repeated for five runs with different random seeds. We denote our ResNet with more skip connections as ResNet (AugSkip). As shown in Table 1, ZerO achieves state-of-the-art accuracy on both datasets. Although we observe that ZerO is slightly worse than standard ResNet with Kaiming initialization by 0.1%, we note that *the degradation is induced by the differences in architectures instead of ZerO initialization itself*. When comparing initialization schemes under the same ResNet (AugSkip), ZerO matches or even beats other random initialization methods.

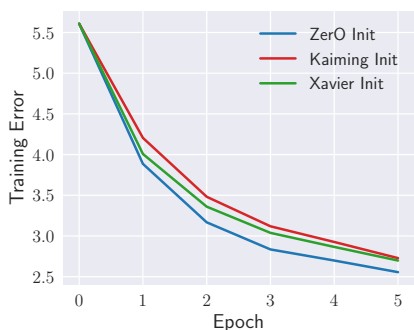

Figure 5: Training error over the first 5 epochs on ImageNet.

| Dataset | Model | Initialization | Test Error (mean $\pm$ std) |
|---|---|---|---|
| CIFAR-10 | ResNet-18 (AugSkip) | **ZerO Init** *(fully-deterministc)* | $5.26 \pm 0.10$ |
| | ResNet-18 | Kaiming Init | $5.14 \pm 0.14$ |
| | ResNet-18 (AugSkip) | Kaiming Init | $5.27 \pm 0.13$ |
| | ResNet-18 | Xavier Init | $5.20 \pm 0.14$ |
| | ResNet-18 (AugSkip) | Xavier Init | $5.28 \pm 0.17$ |
| ImageNet | ResNet-50 (AugSkip) | **ZerO Init** *(fully-deterministc)* | $23.63 \pm 0.04$ |
| | ResNet-50 | Kaiming Init | $23.46 \pm 0.07$ |
| | ResNet-50 (AugSkip) | Kaiming Init | $23.65 \pm 0.09$ |
| | ResNet-50 | Xavier Init | $23.65 \pm 0.11$ |
| | ResNet-50 (AugSkip) | Xavier Init | $23.72 \pm 0.08$ |

Table 1: Benchmarking ZerO on CIFAR-10 and ImageNet. ResNet-50(AugSkip) is our proposed network. We repeat each run 10 times with different random seeds.

**Improved reproducibility.** In addition, ZerO achieves the lowest standard deviation over the repeated runs. On ImageNet, the gap between ZerO and other methods is even more than 40%. Thus, removing the randomness in the weight initialization improves reproducibility, with possible implications for topics such as trustworthy machine learning.

### 4.1 TRAINING WITHOUT NORMALIZATION

We also evaluate the ability of ZerO to remove batch normalization during training. We use the residual blocks that replace batch normalization, as illustrated in Figure 1. We compare various initialization methods using ResNet-50 on ImageNet, including a recent normalization-free method called Fixup (Zhang et al., a). As the training without batch normalization usually suffers from overfitting, we apply a stronger regularization technique Mixup Zhang et al. (b), which is the same as the training setting in (Zhang et al., a). The Mixup coefficient is tuned for each setting, and we train ResNet-50 for 90 epochs. We find that training with ZerO is stable with default hyperparameters such as a learning rate of $0.1$. This is not always the case for training without normalization, as the training with Kaiming initialization fails when using $0.1$ as the learning rate.

As shown in Table 2, without batch normalization, ZerO achieves significantly better results than standard random initialization such as Xavier and Kaiming. We observe that ZerO is slightly worse

than Fixup, which is likely due to the difference in the speed of convergence. When we train both models for 180 epochs, the gap is largely reduced from 0.7% to 0.2%.

| Method | Batch Normalization | Large Learning Rate | Test Error (mean ± std) |
|---|---|---|---|
| Kaiming Init | ✓ | ✓ | $23.46 \pm 0.07$ |
| Kaiming Init + Mixup | ✓ | ✓ | $23.16 \pm 0.09$ |
| *(Mixup enabled)* | | | |
| **ZerO Init** | ✗ | ✓ | $24.52 \pm 0.06$ |
| Fixup Init | ✗ | ✓ | $23.85 \pm 0.10$ |
| Kaiming Init | ✗ | ✗ | $29.91 \pm 0.12$ |
| Xavier Init | ✗ | ✓ | $25.89 \pm 0.11$ |

Table 2: Benchmarking ResNet-50 without batch normalization on ImageNet

## 5 RELATED WORKS

**Theoretical analysis of deep networks**   To ensure stable training with random weight initialization, previous works such as Glorot and Bengio; He et al. (a) study the propagation of variance in the forward and backward pass under different activations. For residual networks, by analyzing the optimization landscape of linear residual networks, Hardt and Ma suggests that all critical points in a neighborhood around zero are proved to be global minima, suggesting zero initialization should be a better choice from the optimization perspective.

**Initialization for ResNet**   Various studies are proposed to address the initialization problem in ResNet (Bachlechner et al.; Gehring et al.; Balduzzi et al.). As the success of batch normalization (Ioffe and Szegedy), De and Smith; Hoffer et al. studied the effect of it, and Zhang et al. (a); De and Smith studied how to train residual networks without batch normalization. Also, Hardt and Ma; Srivastava et al.; Goyal et al.; Zhang et al. (a); Bachlechner et al. found that initializing the residual branches at (or close to) zero benefits the optimization but they still require random initialization. We take this idea to the extreme by initializing all the weights to zero and one.

In another related work, Blumenfeld et al. discusses whether random initialization is needed from the perspective of feature diversity (Rahimi and Recht). They propose networks with identical features at initialization but they still require random noise to improve the performance.

## 6 CONCLUSION

In this work, we propose a fully deterministic initialization ZerO by using skip connections and Hadamard transforms to modify the standard ResNet architecture. Extensive experiments demonstrate that ZerO achieves state-of-the-art performance, suggesting that random weight initialization may not be necessary for modern network initialization.

Our deterministic ZerO initialization opens up many new possibilities. Theorists may be interested in why an over-parameterized neural network with deterministic initial weights achieves such good generalization. Practitioners may apply ZerO to train networks without normalization or train sparse neural networks with pruning techniques. We hope that our results will inspire other researchers to consider deterministic initialization schemes and to rethink the role of weight initialization in training deep neural networks.

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

# A ADDITIONAL DETAILS OF SECTION 2

## A.1 DETAILS OF THE EXPERIMENTS ON MNIST

We construct a simple network based on the one at Figure 2, where we add a linear mapping after the second layer to match the dimension of the labels. The network is: $\mathcal{F}(\boldsymbol{x}) = \boldsymbol{W}^3 \circ \text{Relu}(\boldsymbol{W}^2 + \boldsymbol{I}^2) \circ \text{Relu}(\boldsymbol{W}^1 + \boldsymbol{I}^1)\boldsymbol{x}$ where $\boldsymbol{x} \in \mathbb{R}^{784}$ and $\boldsymbol{y} \in \mathbb{R}^{10}$, which matches the input and label dimensions of the MNIST dataset. Both the first and second layer has 2048 and 2048 hidden spaces, where $\boldsymbol{W}^1, \boldsymbol{I}^1 \in \mathbb{R}^{2048 \times 784}$ and $\boldsymbol{W}^2, \boldsymbol{I}^2 \in \mathbb{R}^{2048 \times 2048}$. $\boldsymbol{W}^3 \in \mathbb{R}^{10 \times 2048}$ is a linear mapping, $\boldsymbol{I}^1$ is a dimension increasing skip connection based on zero padding.

We train the network for 14 epochs using SGD with 0.1 learning rate. We compare with Kaiming Init and Xavier Init by applying their initialization with the standard setting on the example network. All settings achieve the test accuracy above 98% after the training. Our weight distribution plot in the main text is also gathered at the end of the training.

### A.1.1 WEIGHT DISTRIBUTION AT DIFFERENT TRAINING ITERATIONS

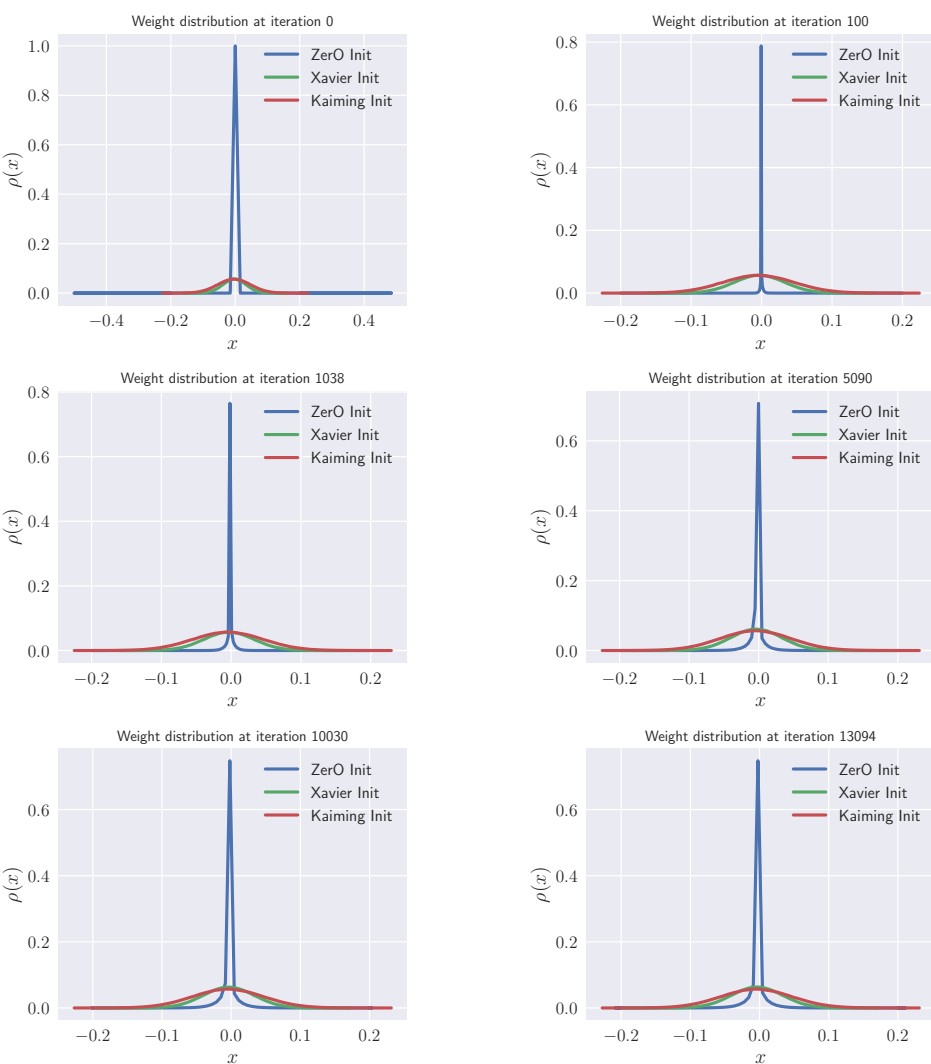

## B    ADDITIONAL DETAILS OF SECTION 3

### B.1    PARTIAL-IDENTITY INITIALIZATION

---

**Algorithm 1** *Partial-Identity Initializer* for $1 \times 1$ convolutional kernel.

---

**Input:**  a $1 \times 1 \times c_{in} \times c_{out}$ convolutional kernel $K$, $c_{in}$ number of input channels, $c_{out}$ number of output channels.
**Step 1.**  For $i = 0, ..., c_{in} - 1$ and $j = 0, ..., c_{out} - 1$, initialize $K[0, 0, i, j] \leftarrow 0$
**Step 2.**  $c_{min} \leftarrow min(c_{in}, c_{out})$
**Step 3.**  For $i = 0, ..., c_{min} - 1$, initialize $K[0, 0, i, i] \leftarrow 1$
**Return**  $K$

---

## C    NETWORK PRUNING

As shown in Figure 3, the network trained by ZerO has a weight variance significantly smaller than the networks trained by random initialization methods. Because the weights that are close to zero usually have smaller effects on the network, they can be pruned to increase the network sparsity while preserving the prediction performance. For ZerO initialization, as most of the weights are extremely close to zero even after training, this motivates us to discover whether ZerO helps to generate sparse networks through pruning.

To verify our hypothesis, we apply a standard magnitude-based pruning for the trained networks initialized with different initializers. We then evaluate the test accuracy of the pruned networks. The magnitude-based pruning method prunes a portion of weights with the lowest magnitudes in each layer. We use the network described in Appendix A for MNIST and ResNet-18 for CIFAR-10. Standard ResNet-18 is adopted for Kaiming and Xavier initializers for better accuracy, and ResNet-18 (AugSkip) is applied for ZerO initializer.

As shown in Figure 6, compared to Kaiming and Xavier initializers, the networks trained with ZerO initializer can be pruned more aggressively while preserving the test accuracy. Because the state-of-the-art pruning methods adopt standard random initializers by default, we believe ZerO would be a powerful replacement that improves the pruning performance.

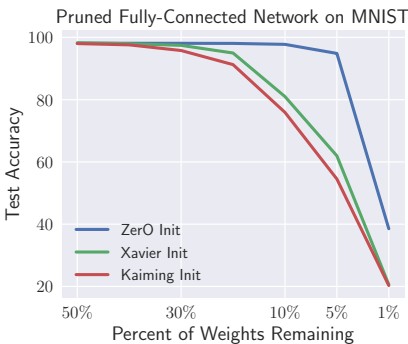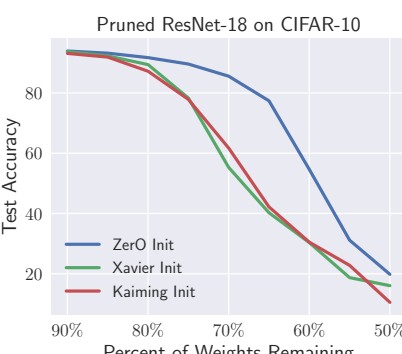

Figure 6: Test accuracy of the pruned networks trained with various initialization methods.

# D  ADDITIONAL ABLATION STUDIES

## D.1  BENCHMARKING HADAMARD TRANSFORM

In this ablation study, we evaluate the effects of our modifications to the original ResNet, including 1) adding additional skip connections and 2) applying Hadamard transform. As shown in Table 3, we evaluate different configurations using ResNet-18 on CIFAR-10. The results suggest that both additional skip connections and Hadamard transforms are needed for ZerO initialization. This verifies that the problems we discussed in Section 2 are crucial for initializing the weights to zero.

In addition, the results indicate that the slight accuracy degradation comes from the Hadamard transform instead of the additional skip connections. This motivates us to discover a better way to break the training degeneracy while preserving the performance in the future.

| Additional skips | Hadamard transform | ZerO Init | Kaiming Init | Xavier Init |
|:---:|:---:|:---:|:---:|:---:|
| ✗ | ✗ | 89.24 | 5.14 | 5.20 |
| ✗ | ✓ | 87.52 | 5.24 | 5.26 |
| ✓ | ✗ | 47.26 | 5.14 | 5.20 |
| ✓ | ✓ | 5.26 | 5.27 | 5.28 |

Table 3: The effects of additional skip connections and Hadamard transform in ResNet-18 on CIFAR-10. ResNet-18 without both additional skips and Hadamard transform is the standard ResNet-18. Test errors (%) are reported.

## D.2  LEARNING RATE WARMUP

We observe that learning rate warmup is essential for ZerO to achieve a large maximal learning rate. This is because a large initial learning rate leads to gradient explosion, as most of the weights start from the exact zero at the beginning. To demonstrate it, we measure the gradient norms during training for the settings with or without learning rate warmup.

As shown in Figure 7, we present the gradient norms wrt. the weights of the first convolutional layer in ResNet-18, which is trained on CIFAR-10 for the first 400 iterations. The results indicate that learning warmup is needed for ZerO because the smaller initial learning rate prevents the gradients from explosion at the beginning of training.

Interestingly, we observe that the warmup has much larger effect on ZerO than other random initialization methods. For Kaiming and Xavier initialization, gradient norms are nearly unchanged after applying warmup. However, for ZerO initialization, the gradient norms become significantly smaller (even smaller than the baselines) after applying the warmup.

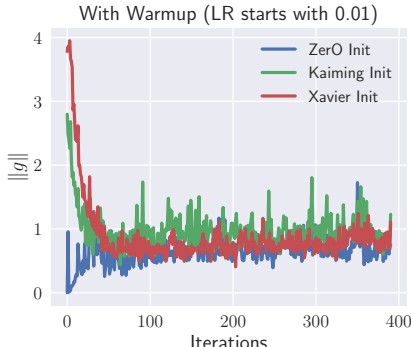 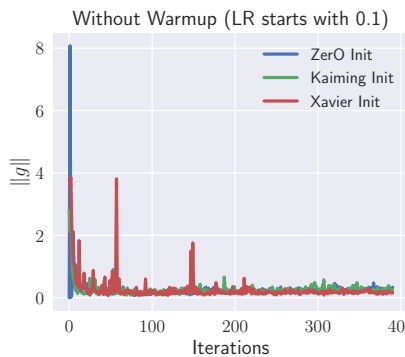

Figure 7: Measuring gradient norms of ResNet-18 for the first 400 iterations.

For random initialization methods, the small changes of gradient norms are because the gradient norms are largely dominated by the initial weight variance, which are controlled by the initializers.

Choosing inappropriate initial weight variance usually leads to gradient explosion. To avoid the explosion, previous studies on random initialization propose various principles to control the initial weight variances, given different architectures and types of nonlinearities. However, as ZerO initialization trains the weights from zeros and ones without pre-defined weight variances, the learning rate becomes the dominant factor that controls the norm of the initial gradients. Hence, choosing an appropriate initial learning rate is sufficient for avoiding exploding gradients under ZerO initialization, without the need of considering the variances of the initial weights.

