# OpenReview forum: "ZerO Initialization: Initializing Residual Networks with only Zeros and Ones"
_ICLR.cc/2022/Conference — ICLR 2022 Submitted_

### Official Review · Reviewer_RE44 · 2021-10-19

**Correctness:** 4
**Technical Novelty And Significance:** 2
**Empirical Novelty And Significance:** 2
**Recommendation:** 5
**Confidence:** 3

**Main Review:**

Strengths:
1. It is interesting to show that ResNets can be trained to get competitive performance with zero-and-one initialized weights. The conventional view is that the weights should be initialized with random values for symmetry-breaking.
2. The problems (e.g. dead neuron) discussed in the paper are well-illustrated. In Section 2.1, the authors use simple and clear examples to demonstrate how zero-initialized weights can cause training to get stuck and why short connections can rescue. In Section 2.2-2.3, the authors explain in a similar manner why Hadamard transform can deal with degeneracy problem caused by the dimensionality expansion.
3. Experimental results confirm that randomness can be removed from weight initialization. The authors compare several initialization schemes on the standard ResNet models on CIFAR-10 and ImageNet and show theirs achieves competitive results in terms of convergence speed and test accuracy.

Weaknesses:
1. The motivation of the zero-and-one initialization is unclear. What are the theoretical advantages of removing randomness in initialization? It seems to me that keeping randomness in initialization has the advantage of potentially avoiding bad local minima and saddle point. Empirically, despite the authors have shown some comparison with He and Xavier methods, the zero-and-one initialization does not show *significant* advantages.
2. The zero-and-one initialization is of a heuristic nature and lacking of theoretical background. There is no unique way to avoid the dead neuron problem. For a trivial example, one can simply add a shortcut connection from the dead neuron to a output neuron and one from a input neuron to the dead neuron. It would be desirable to motivate the proposed initialization scheme with some theoretical optimality.
3. Additional costs of the zero-and-one initialization. The extra shortcut connections and Hadamard transform result in additional computational cost and model complexity. As they are cheap operations, I consider this issue a minor one.
4. In Section 1, the authors state "In order to design networks robust to vanishing gradients at initialization, He et al. introduced
skip connections". However, it is contradictory to the claim in the original paper of He et al, which states "We argue that this optimization difficulty is unlikely to be caused by vanishing gradients."

**Summary Of The Paper:**

This paper proposes a new initialization scheme for ResNet. The key feature is the initialization values are set to all zeros and ones, rather than random. The development of this initialization scheme is motivated by avoiding the "dead neuron" problem and "training degeneracy" problem for networks with zero-initialized weights. Some toy examples are discussed to illustrate the motivation. Last, experiments on standard benchmarks are conducted to verify the initialization scheme.

**Summary Of The Review:**

The paper is technically correct but lack of strong motivation, theoretical advances and empirical performance.

---

> ### Author Response · Authors · 2021-11-19
> **Response to reviewer RE44**
>
> Thanks for your thoughtful feedback. We will address your questions and suggestions in the order of your review.
>
> > The motivation of the zero-and-one initialization is unclear. What are the theoretical advantages of removing randomness in initialization? It seems to me that keeping randomness in initialization has the advantage of potentially avoiding bad local minima and saddle point. Empirically, despite the authors have shown some comparison with He and Xavier methods, the zero-and-one initialization does not show significant advantages.
>
> As presented in [1], for a deep linear residual network, its objective function has an optimal solution with a fairly small norm, which represents the residual network close to identity mapping. In addition, [1] proves that all critical points are global minima for a small region that includes the origin. This suggests that when initializing the weights at the origin, the network converges to the global minima more easily, compared to other random initializations that start from a point that deviates from the origin.
>
> Our empirical results support this hypothesis. As shown in Table 1 in the paper, ZerO initialization achieves lower test error compared to Xavier and Kaiming random initialization under ResNet (AugSkip), which resembles the “identity” network assumed in [1]. In addition, according to Figure 3 (Left), the final weight distribution trained with ZerO has a significantly smaller weight variance, suggesting the norm of the final solution is very small. This suggests the solution is close to the identity mapping, which supports the theoretical conclusions in [1].
>
> Compared to random initialization, we believe that stochastic gradient descent plays a more important role in avoiding bad local minima and saddle points. The training with ZerO initialization still has randomness in batch selection so the benefits of SGD are preserved.
>
> Regarding empirical evaluations, we believe the results of ZerO initialization are significant. First, we think ZerO is the first deterministic initialization for deep neural networks that achieves SOTA performance over various tasks. In addition, compared to Kaiming and Xavier methods, ZerO has significant advantages over normalization-free training, model reproducibility, and network pruning (network pruning are the new results we presented in Appendix C).
>
>
> [1] Hardt, Moritz, and Tengyu Ma. "Identity matters in deep learning." arXiv preprint     arXiv:1611.04231 (2016).
>
> > The zero-and-one initialization is of a heuristic nature and lacking of theoretical background. There is no unique way to avoid the dead neuron problem. For a trivial example, one can simply add a shortcut connection from the dead neuron to a output neuron and one from a input neuron to the dead neuron. It would be desirable to motivate the proposed initialization scheme with some theoretical optimality.
>
> Thanks for your suggestions, we agree that some theoretical backgrounds would improve the readability, and we will include some as motivations. The dead neuron problem we discussed is limited to the situation when the weights are initialized as zero. Thus, adding shortcut connections would be the desired way to solve the problem.
>
> > Additional costs of the zero-and-one initialization. The extra shortcut connections and Hadamard transform result in additional computational cost and model complexity. As they are cheap operations, I consider this issue a minor one.
>
> That’s correct. We rigorously profile the number of MACs needed for ResNet-50 and ResNet-50 (AugSkip). For a single forward propagation, ResNet-50 needs 4.11 GMAC and ResNet-50 (AugSkip) needs 4.23 GMAC, where the structures with additional skip connections only introduce 2.9% computational overhead.
>
> Also, thanks for your other suggestions regarding clarity in the introduction. We will modify our statement in the revision.

---

> > ### Comment · Reviewer_RE44 · 2021-11-30
> > **Thank you for the response**
> >
> > I have read the authors' response. While the authors are making an effort to make their work more complete, I still find the work lack of theoretical ground and the empirical findings are not strong enough. Besides, the authors didn't address point 4 in the weaknesses of my review. Therefore, I keep my original rating.

---

> > > ### Author Response · Authors · 2021-11-30
> > > **Thank you for the reply**
> > >
> > > Thanks for your reply. For point 4, we agree with you that avoiding vanishing gradients is not the main objective of the ResNet introduced by He et al [1], which is because batch normalizations alleviate the vanishing gradient problem for both plain and residual networks. However, skip connections do help avoid the vanishing gradients as the gradients are accumulated through skip connections during backward propagation. Thus, skip connections play a more important role for networks without batch normalizations. As mentioned in He et al [1], the studies of [2, 3] design networks (without BN) that have a few intermediate layers are directly connected to auxiliary classiﬁers for addressing vanishing gradients. For our paper, we will correct our citation by pointing out that skip connections avoid vanishing gradients especially for networks without batch normalization.
> > >
> > > [1] He, Kaiming, et al. "Identity mappings in deep residual networks." European conference on computer vision. Springer, Cham, 2016.
> > >
> > > [2] Szegedy, Christian, et al. "Going deeper with convolutions." Proceedings of the IEEE conference on computer vision and pattern recognition. 2015.
> > >
> > > [3] Lee, Chen-Yu, et al. "Deeply-supervised nets." Artificial intelligence and statistics. PMLR, 2015.

---

### Official Review · Reviewer_9bJj · 2021-11-02

**Correctness:** 4
**Technical Novelty And Significance:** 3
**Empirical Novelty And Significance:** 3
**Recommendation:** 6
**Confidence:** 3

**Main Review:**

Strength:
The idea of ZerO is novel and has various benefits such as improving the reproducibility of models and training networks without batch normalization.

Weakness:
1. One drawback of ZerO is that it can only be applied to residual networks.
2. Can the authors also provide a comparison with the baselines for space-time complexity? I think that the number of skip connections added is significant which may add to the complexity.
3. How about applying ZerO for other core CV applications such as segmentation and object detection. Does the behavior of ZerO remain the same for such applications?

**Summary Of The Paper:**

The paper proposes a deterministic weight initialization scheme called ZerO for residual networks.  ZerO works by first augmenting existing ResNet architectures with extra skip connections and Hadamard transforms and then initializes network weights with only zeros and
ones (instead of random weights). The authors show that the proposed scheme has various benefits such as improving reproducibility, training networks without batch normalization, and improved performance.

**Summary Of The Review:**

Overall the paper presents some interesting ideas. However, there are some concerns such as the space-time complexity of models and limited applicability.

---

> ### Author Response · Authors · 2021-11-19
> **Response to reviewer 9bJj**
>
> Thanks for your comments. We will address your questions and suggestions in the order of your review.
>
> > One drawback of ZerO is that it can only be applied to residual networks.
>
> ZerO can be applied to other architectures as long as they follow our design principles to solve the dead neuron and training degeneracy problems discussed in the paper. We propose ResNet (AugSkip) that satisfies the design principles as an illustration to show the benefits of ZerO initialization in practice. We believe ZerO can also be applied to other architectures such as transformers.
>
> > Can the authors also provide a comparison with the baselines for space-time complexity? I think that the number of skip connections added is significant which may add to the complexity.
>
> The additional overhead is not significant. We rigorously profile the number of MACs needed for ResNet-50 and ResNet-50 (AugSkip). For a single forward propagation, ResNet-50 needs 4.11 GMAC and ResNet-50 (AugSkip) needs 4.23 GMAC, where the structures with additional skip connections only introduce 2.9% computational overhead.
>
> > How about applying ZerO for other core CV applications such as segmentation and object detection. Does the behavior of ZerO remain the same for such applications?
>
> We believe ZerO would work well for other applications as long as their architectures satisfy our design principles. We appreciate your suggestions, and we will benchmark ZerO to other CV applications in the future.
>
> In addition, we conducted experiments on another application of network pruning during the rebuttal. We discussed the experiment results in Appendix C. The results show that the networks trained with ZerO can be pruned into much smaller models while preserving the test accuracy.

---

### Official Review · Reviewer_8Znp · 2021-11-02

**Correctness:** 2
**Technical Novelty And Significance:** 3
**Empirical Novelty And Significance:** 3
**Recommendation:** 5
**Confidence:** 3

**Main Review:**

Pros:
- This paper is easy to follow.
- The proposed method is interesting and somewhat novel.

Cons:
- This paper seems to propose a new structure (containing extra skip connections and Hadamard transforms) that contradicts the topic ( an initialization). The proposed method ZerO overly relies on auxiliary structures, compared with FixUp [1] that only uses a multiplier and bias. Therefore, the proposed method is not a pure initialization for ResNet.
- In Table 2, does ZerO only use a structure with AugSkip? Wether other initializations (i.e., Fixup, Kaiming, and Xavier) choose vanilla ResNet? There is no description to mention it. In addition, ZerO performs worse than Fixup. Although it can reduce the gap by running 180 epochs, it costs too much on time.
- It is suggested to change notation like $W^2$ to $W^{(2)}$.


Additional Questions:
- As a fully-deterministic structure, the initial weights are the same with different seeds. Why is the std not equal to 0? What else makes an effect on the training?
- Will the auxiliary structures influence time consumption?
- Kaiming initialization is proposed for relu activation function and ResNet adopts relu. Therefore, is it weird that Kaiming is worse than Xavier in Table 2?

[1] Hongyi Zhang, Yann N. Dauphin, Tengyu Ma. Fixup Initialization: Residual Learning Without Normalization. ICLR 2019.

**Summary Of The Paper:**

To initialize a ResNet with only zeros and ones, this paper analyses the problem of all-zero initialization and then proposes a  method named ZerO to achieve the goal by augmenting the standard ResNet architectures with a few extra skip connections and Hadamard transforms. The proposed idea is interesting and somewhat novel. However, it seems to introduce a new structure rather than only initialization, and empirical results are not really good.

**Summary Of The Review:**

Although the idea is somewhat novel, for the unremarkable empirical results, incongruent method contradicting the topic initialization, I regret to reject it weakly.

---

> ### Author Response · Authors · 2021-11-19
> **Response to reviewer 8Znp**
>
> Thank you for reviewing our paper. We will address your questions and suggestions in the order of your review.
>
> > This paper seems to propose a new structure (containing extra skip connections and Hadamard transforms) that contradicts the topic ( an initialization). The proposed method ZerO overly relies on auxiliary structures, compared with FixUp [1] that only uses a multiplier and bias. Therefore, the proposed method is not a pure initialization for ResNet.
>
> The main objective of the paper is to discuss the difficulties and potential benefits when initializing residual networks with only zeros and ones. While Fixup is a method based on Kaiming random initialization targeting the normalization-free training, ZerO is a new deterministic initialization outside the family of random initialization. Thus, we believe that ZerO and Fixup are not comparable in terms of design simplicity.
>
> On the other hand, unlike Fixup, ZerO does not need to design sophisticated principles for controlling weight variances, such as scaling the weight layers by L^{-1/2m-2}. This is because ZerO doesn’t need pre-defined weight variances. In addition, unlike Fixup, ZerO has various benefits except normalization-free training, such as its benefits for reproducibility and network pruning, where network pruning is the new results we presented in Appendix C.
>
> > In Table 2, does ZerO only use a structure with AugSkip? Wether other initializations (i.e., Fixup, Kaiming, and Xavier) choose vanilla ResNet? There is no description to mention it. In addition, ZerO performs worse than Fixup. Although it can reduce the gap by running 180 epochs, it costs too much on time.
>
> We apologize for the confusion. ZerO is trained with ResNet (AugSkip) in Table 2. Kaiming and Xavier choose standard ResNet, and Fixup has the modified ResNet. We will add the description to the paper. Although ZerO performs worse than Fixup under normalization-free settings, we want to reiterate that ZerO is not solely designed for normalization-free settings. Fixup does not provide the other benefits achieved by ZerO.
>
> > As a fully-deterministic structure, the initial weights are the same with different seeds. Why is the std not equal to 0? What else makes an effect on the training?
>
> The batch selection process for training networks with SGD is still random. Thus, the networks trained by SGD still converge to different solutions given different random seeds.
>
> > Will the auxiliary structures influence time consumption?
>
> The auxiliary structures only incur a small amount of time overhead. As discussed in the paper, because a Hadamard matrix only consists of -1 and 1, applying a Hadamard matrix does not require any multiplication as sign flips are all it takes. It can be implemented efficiently with only O(nlog(n)) complexity.
>
> In addition, we rigorously profile the number of MACs needed for ResNet-50 and ResNet-50 (AugSkip). ResNet-50 needs 4.11 GMAC and ResNet-50 (AugSkip) needs 4.23 GMAC for a single forward propagation, where the auxiliary structures only introduce 2.9% computational overhead.
>
> > Kaiming initialization is proposed for relu activation function and ResNet adopts relu. Therefore, is it weird that Kaiming is worse than Xavier in Table 2?
>
> In Table 1, Kaiming Init is better than Xavier Init when batch normalization is enabled. This matches the results in the original paper proposing pre-activation ResNet [1], where batch normalization is enabled as well. For normalization-free settings, the pre-activation ResNet with Kaiming Init can not tolerate the large learning rate of 0.1, which causes a large accuracy gap between them.
>
> [1] He, Kaiming, et al. "Identity mappings in deep residual networks." European conference on computer vision. Springer, Cham, 2016.

---

### Official Review · Reviewer_P5yd · 2021-11-03

**Correctness:** 3
**Technical Novelty And Significance:** 3
**Empirical Novelty And Significance:** 2
**Recommendation:** 5
**Confidence:** 4

**Main Review:**

### Strength:

#### 1. The motivation of the paper is clear and sound.

#### 2. The idea of combining Hadamard transform with identity mappings to initialize ResNet-style architectures is not previously seen in the literature.

#### 3. Deterministic initialization reduces source of randomness, as is shown by the standard errors of trial results of different methods.

#### 4. The paper is easy to comprehend.

#### 5. Code is shared anonymously for reproducibility.


### Weakness:

#### 1. While the approach as a whole is new to my best knowledge, most ideas have been investigated in published work. For example, identity initialization of convolutional kernels was presented in Xiao et. al. (2018) under the name of Delta-Orthogonal initialization; additional residual connections appear in related work such as Hardt and Ma (2017); Hadamard transform layer is used in binary neural networks, among other neural network applications.

#### 2. The experimental results do not show performance advantage of the proposed method over previous ones such as the original ResNet and the Fixup initialization (in the no normalization setting).

#### 3. The mathematical / technical contributions are limited; though this may have the virtue of improving the overall readability.

*Reference:*

*Xiao et. al. (2018). Dynamical Isometry and a Mean Field Theory of CNNs: How to Train 10,000-Layer Vanilla Convolutional Neural Networks.*

*Hardt and Ma (2017). Identity Matters in Deep Learning.*

### Other comments and concerns:

#### 1. I am interested in the final weight distributions shown in Figure 3 (Left). I wonder whether this figure implies the proposed method can obtain solutions "closer" to identity mapping.

##### 1.1. Do the plots correspond to the same network architecture, only different initializations?

##### 1.2 What are the initial weight variances of different initializations (as it could be that they start from very different distributions)?

##### 1.3. Since the network block contains a scalar multiplier, there can be identical blocks (in terms of the input-output mapping) with very different weight distributions. Could that be the case here?

##### 1.4. Could you provide more justification as to the connection of close-to-zero weight distribution to weight pruning? In my understanding, a more direct argument is to analyze the network's sensitivity to weight perturbation; due to the presence of scalar multipliers and/or normalization layers, small weight scale does not necessarily imply small sensitivity.


#### 2. In "Experiments - Hyperparameter settings", you mention "learning rate warmup is essential for ZerO to achieve a large maximal learning rate". This seems to suggest that the proposed initialization still suffers from training instability resulted from gradient explosion.
##### 2.1. To address this concern, could you provide some comparison (e.g. gradient norm plots) of your method vs. the baselines in the first hundreds of steps with and without warmup schedule?

**Summary Of The Paper:**

In this paper, the authors propose some modifications to the classic ResNet architecture and initialization scheme, so that the network can be trained with fully deterministic initialization, even without any normalization layers. The authors motivate their design choices by identifying and analyzing the problems with the forward and backward dynamics of some naive ideas. Then the authors present their approach and explain why each choice they make can solve the specific problems they have presented. Then they evaluate their proposed method along with some baselines on two popular image classification datasets, showing that their deterministic initialization method is on par with the ones with random weight initialization. Finally, they present related work and conclusion.

**Summary Of The Review:**

The paper introduces a novel method to initialize a ResNet-like architecture deterministically with binary weights (except for the Hadamard transform where additional rescaling is required). The paper contains some novelty in its model design, has good clarity, but does not show convincing practical significance compared with previous work. Some claims are hypothetical and may require further justification or be supported by further evidence.

---

> ### Author Response · Authors · 2021-11-19
> **Response to reviewer P5yd (Part 1)**
>
> Thanks for your thoughtful feedback. We will address your questions and suggestions in the order of your review.
>
> > While the approach as a whole is new to my best knowledge, most ideas have been investigated in published work. For example, identity initialization of convolutional kernels was presented in Xiao et. al. (2018) under the name of Delta-Orthogonal initialization; additional residual connections appear in related work such as Hardt and Ma (2017); Hadamard transform layer is used in binary neural networks, among other neural network applications.
>
> We agree that our ZerO initialization is related to these works. However, there are key differences as listed below.
>
> For Delta-Orthogonal initialization, although its identity initialization is similar to ZerO, it still needs to initialize weights randomly using random orthogonal matrices, and sometimes it also needs to tune the scale of the initial weights. On the other hand, ZerO does not need any randomness for weight initialization or a pre-defined initial weight scale.
>
> For the works by Hardt and Ma (2017), they designed an initialization based on random normal initialization with substantially smaller initial weights, and they suggested that such a small scale weight initialization helps the training. However, they couldn’t remove the randomness and reduce the weight scale to exactly zero because their architecture still has the dead neuron and training degeneracy problems as discussed in our paper.
>
> On the other hand, we believe that ZerO supports the theoretical results presented in Hardt and Ma (2017). Their work suggests that the objective function for a deep residual network has an optimal solution with a fairly small norm. Because ZerO initializes the weights extremely close to zero (with 99.9% zero weights at initialization), the ZerO-initialized network should converge to this small norm global solution more easily compared to randomly initialized networks. Our empirical results demonstrate this hypothesis. As shown in Table 1 in the paper, ZerO initialization achieves lower test error compared to Xavier and Kaiming random initialization under ResNet (AugSkip). ResNet (AugSkip) resembles the “identity” network assumed in Hardt and Ma (2017).
>
> The Hadamard transform in our network serves to increase the dimensionality and break the training degeneracy, which differs from its uses cases in other fields.
>
> > The experimental results do not show the performance advantage of the proposed method over previous ones such as the original ResNet and the Fixup initialization (in the no normalization setting).
>
> We would like to point out that ZerO achieves better results than Xavier and Kaiming Init under the same ResNet (AugSkip), suggesting the slight degradation comes from the modified architecture instead of the ZerO initialization itself. We believe our results are significant given that we are the first work that matches SOTA accuracy with deterministically initialized networks. In appendix D.1, we further verify that the slight degradation comes from Hadamard transform instead of adding more skip connections, which motivates us to find alternatives to break the training degeneracy in the future.
>
> In addition, unlike Fixup, ZerO has various benefits except normalization-free training. With its deterministic property, ZerO achieves better reproducibility. With its zero and one property, ZerO helps to generate sparse networks with fewer parameters while preserving accuracy (these are new results presented in Appendix C). Therefore, we believe ZerO achieves empirical significance from various perspectives.
>
> > I am interested in the final weight distributions shown in Figure 3 (Left). I wonder whether this figure implies the proposed method can obtain solutions "closer" to identity mapping.
>
> Yes, we agree. As discussed in previous point 1, ZerO supports the theoretical results presented in Hardt and Ma (2017) as it should converge to the small norm global solution more easily compared to randomly initialized networks. Figure 3 (Left) also demonstrates that the norm of the final solution is very small, suggesting it can obtain solutions ‘closer’ to identity mapping.
>
> > Do the plots correspond to the same network architecture, only different initialization?
>
> Yes, the plots in Figure 3 correspond to the same fully-connected network defined in Appendix A.1. Only the initialization methods are different.

---

> > ### Author Response · Authors · 2021-11-19
> > **Response to reviewer P5yd (Part 2)**
> >
> > > What are the initial weight variances of different initializations (as it could be that they start from very different distributions)?
> >
> > For Kaiming and Xavier initializations, we use the initializers provided by PyTorch, and their initial weight variances follow their default factors without scaling (please refer to https://pytorch.org/docs/stable/nn.init.html).
> > For ZerO initialization, it doesn’t need to define the initial weight variance as every weight starts from zero or one. The initial weight variance of ZerO can be viewed as a factor depending on the learning rate. We discussed its relationship in Appendix D.2 (where we answered your following questions about learning warmup).
> >
> > > Since the network block contains a scalar multiplier, there can be identical blocks (in terms of the input-output mapping) with very different weight distributions. Could that be the case here?
> >
> > We don’t have scalar multipliers for the network in Figure 3. It is just a plain fully-connected network.
> >
> > > Could you provide more justification as to the connection of close-to-zero weight distribution to weight pruning? In my understanding, a more direct argument is to analyze the network's sensitivity to weight perturbation; due to the presence of scalar multipliers and/or normalization layers, small weight scale does not necessarily imply small sensitivity.
> >
> > First of all, we conducted additional experiments on network pruning showing that ZerO has a significant benefit (in Appendix C). We agree that a small weight scale does not imply small sensitivity. However, if we prune a fixed portion of weights within each layer, the small relative weight scale in each layer should imply small sensitivity, which does not depend on the scalar multipliers.
> >
> > > In "Experiments - Hyperparameter settings", you mention "learning rate warmup is essential for ZerO to achieve a large maximal learning rate". This seems to suggest that the proposed initialization still suffers from training instability resulted from gradient explosion. To address this concern, could you provide some comparison (e.g. gradient norm plots) of your method vs. the baselines in the first hundreds of steps with and without warmup schedule?
> >
> > Yes, it is right. Without the learning rate warmup, an inappropriate large initial learning rate will lead to gradient explosion. This is because, unlike random initializations, the gradient norms under ZerO only depend on the initial learning rate instead of the pre-defined weight variances. Therefore, choosing an appropriate initial learning rate is important for ZerO initialization. We discussed this point in Appendix D.2, where we also present the gradient norm plots as you requested.

---

### Official Review · Reviewer_PQDc · 2021-11-07

**Correctness:** 3
**Technical Novelty And Significance:** 1
**Empirical Novelty And Significance:** 2
**Recommendation:** 5
**Confidence:** 4

**Main Review:**

Strengths:
* (Significance) The paper explores common-sense modifications that are important for understanding the operation of residual networks.
* (Significance) I found the results of the paper to be interesting.
* (Clarity) The majority of the paper is clearly written.
* (Reproducibility) The paper provides code, helping reproducibility.
* (Quality) The experiments include multiple seeds for every configuration, with a clear description of the used hyperparameters. The tasks used for experiments are highly relevant (e.g: Resnet50 for Imagenet). As such, I found the results to be highly reliable.

Weaknesses:
* (Significance/ novelty) Getting rid of batch normalization was already achieved by fixup, in a very similar method to what the paper presented. (Which did not require initialization with only ones and zeros). The experiments do not show a benefit for using the new modified resnet over fixup. Other works, like [SkipInit](https://arxiv.org/abs/2002.10444) and [ReZero](https://arxiv.org/abs/2003.04887) also offer similar methods to improve convergence speed and get rid of batch normalization.

* (Significance) The paper pride its initialization on being deterministic, but this is poorly defined. I could argue that my initialization would be just as deterministic by using constant seed or importing weights saved on storage.

* (Quality- Empirical) I was not convinced that the 4th contribution (lower standard deviation) is correct. Comparing standard deviation over 5 seeds does not give you a statistically significant result, this could easily be the result of random chance. I was also not convinced this is important: If my goal was to reproduce my run, I wouldn't have changed the seed.

* (Novelty) Initializing residual networks to only zeros and constant was already done in [this work](https://arxiv.org/abs/2007.01038). This paper cites this work as requiring "random noise to improve the performance". However, the prior work also concluded that "standard GPU operations, which are non-deterministic, can serve as a sufficient source of symmetry breaking to enable training" (quote from the abstract). Given that this paper also uses GPUs and did not mention using them on deterministic mode (It's stochastic by default), the specified distinction between this work and the prior work is not valid.

* (Clarity) I was confused by section 2.2. As far as I can tell, the degeneracy problem there is a direct result of the choice of padding $\mathbb{I}_{1,2}$ with zeros when the network is wide. Since this is not a trivial solution to the expanding width problem, the title "Width expansion leads to training degeneracy" is misleading, and the addressed problem in this section is poorly defined. I am also confused by why this problem is different from the first dead neuron problem (Skip connections obviously can't solve dead-neuron problems for neurons it is not applied on).

* (Clarity) Section 2.4:  my understanding was that training degeneracy refers to a problem where padded zero values cause the gradient to remain zero  (section 2.2). I do not understand why $C_f$ and $C_b$ are good measures for that kind of degeneracy. Also, $C_f$ and $C_b$ aren't properly defined well for zero-initialized weights.

* (Quality) The paper concludes with results I consider to be negative. I thought this was interesting, as some of the changes made in this paper seem trivial and I did not expect them to harm performance. However, there was no proper ablation study (Figure 3 helped, but wasn't sufficient), so it remains unclear what element exactly caused the small degradation in performance. For example, I wonder if the single-convolution-skip architecture is responsible, but there is no way to disentangle the results from the entire zero-initialized-hadamard network.

**Summary Of The Paper:**

The paper suggests an adjustment to the resnet architecture, which enables successful training despite having deterministically initialized weights.

**Summary Of The Review:**

The paper suggests some interesting modifications to the resnet architecture. However, the changes aren't very novel, and the paper did not convince me that these new adjustments are beneficial. The results in the paper appear to be (slightly) negative, but there is no proper ablation study for them to be informative

---

> ### Author Response · Authors · 2021-11-19
> **Response to reviewer PQDc (Part 1)**
>
> We want to thank the reviewer for your thoughtful feedback. We will address your questions and suggestions in the order of your review.
>
> > (Significance/ novelty) Getting rid of batch normalization was already achieved by fixup, in a very similar method to what the paper presented. (Which did not require initialization with only ones and zeros). The experiments do not show a benefit for using the new modified resnet over fixup. Other works, like SkipInit and ReZero also offer similar methods to improve convergence speed and get rid of batch normalization.
>
> Our initialization differs significantly from Fixup because Fixup still relies heavily on random initialization. As mentioned in their paper, they suggest that “rule 2”, which initializes layers using standard Kaiming initialization with their proposed weight scaling, is the essential part of the Fixup initialization. On the other hand, our ZerO initialization does not require random weights or a sophisticated design for weight scaling.
>
> In addition, the main objective of our paper is to demonstrate that a deterministic initialization works as well as random initialization, and ZerO is the first deterministic method achieving it. Unlike Fixup, SkipInit, or ReZero, we show that ZerO is a versatile initialization that has multiple benefits such as training without batch normalization, enhancing reproducibility, and achieving better network pruning (which is our new result discussed in Appendix C). Therefore, we believe that there is a significant novelty.
>
>
>
> > (Significance) The paper pride its initialization on being deterministic, but this is poorly defined. I could argue that my initialization would be just as deterministic by using constant seed or importing weights saved on storage.
>
> We agree that the question of what it means for an algorithm to be deterministic is a subtle and somewhat subjective question.
> For our purposes, deterministic means that we fix the values of initialization of our weights and they do not involve random variables. The main difference from selecting a constant seed or importing weights saved on storage is that those still need a careful design of initial variances for weights, which is especially challenging in deeper networks. We alleviate that with our use of deterministic and fixed choice of initialization.
> More broadly, the standard definition of randomized algorithms is that we could have drawn the randomness beforehand and provided it as an input to the algorithm, which aligns with our usage of the term.
>
> As you remark, this definition is not fully objective, as we could have included the implementation of a pseudorandom generator in our algorithm and asked the user to provide a single integer serving as seed, as an input. However, such an algorithm would be extremely contrived whereas ZerO follows a small number of design principles.
>
>
>
> > (Quality- Empirical) I was not convinced that the 4th contribution (lower standard deviation) is correct. Comparing standard deviation over 5 seeds does not give you a statistically significant result, this could easily be the result of random chance. I was also not convinced this is important: If my goal was to reproduce my run, I wouldn't have changed the seed.
>
> We have conducted 5 more repeated experiments and updated the results in the revised paper. According to the latest result over 10 repeated runs, ZerO still consistently achieves smaller standard deviations than random methods.
>
> Fixing a random seed is not the solution to reproducibility. This is because: if the findings depend on a particular choice of random seeds, the findings (or results) are not reproducible even if the particular program execution is. Section 2 of the paper [1] well explains this problem, and here we quote a paragraph of their conclusions:
>
> *"What we would like to point out here, is that there are two forms of reproducibility that can interfere if we are not cautious. The reproduction of the results requires the conversion of a stochastic system into a deterministic one, e.g.the seeding process. While this helps the reproduction of results, avoiding this source of variation altogether in experiments has the potential effect of dramatically weakening the generality of conclusions. This is at odds with the reproduction of findings."*
>
> [1] Bouthillier, X., Laurent, C. &amp; Vincent, P.. (2019). Unreproducible Research is Reproducible. Proceedings of the 36th International Conference on Machine Learning, in Proceedings of Machine Learning Research 97:725-734

---

> > ### Author Response · Authors · 2021-11-19
> > **Response to reviewer PQDc (Part 2)**
> >
> > > (Novelty) Initializing residual networks to only zeros and constant was already done in this work. This paper cites this work as requiring "random noise to improve the performance". However, the prior work also concluded that "standard GPU operations, which are non-deterministic, can serve as a sufficient source of symmetry breaking to enable training" (quote from the abstract). Given that this paper also uses GPUs and did not mention using them on deterministic mode (It's stochastic by default), the specified distinction between this work and the prior work is not valid.
> >
> > There are considerable differences between those two works. First, we enabled deterministic GPU operations for all experiments, suggesting random noise, even from stochastic GPU operations, is not necessary to enable training. Second, the methodologies are different. The previous work [1] tries to ensure that each layer receives identical features at initialization, where the inputs of the network are averaged to remove the diversity of inputs. However, our ZerO initialization focuses on propagating input diversity through every layer instead of eliminating such diversity, which we believe is the key to training networks without random initial weights.
> >
> > [1] https://arxiv.org/abs/2007.01038
> >
> > >  (Clarity) I was confused by section 2.2. As far as I can tell, the degeneracy problem there is a direct result of the choice of padding I1,2 with zeros when the network is wide. Since this is not a trivial solution to the expanding width problem, the title "Width expansion leads to training degeneracy" is misleading, and the addressed problem in this section is poorly defined. I am also confused by why this problem is different from the first dead neuron problem (Skip connections obviously can't solve dead-neuron problems for neurons it is not applied on).
> >
> > >  (Clarity) Section 2.4: my understanding was that training degeneracy refers to a problem where padded zero values cause the gradient to remain zero (section 2.2). I do not understand why Cf and Cb are good measures for that kind of degeneracy. Also, Cf and Cb aren't properly defined well for zero-initialized weights.
> >
> > We would like to answer your questions about clarity together as they are related.
> >
> > The first and second problems are similar but different under certain perspectives. For the first dead neuron problem, it can be solved by simply applying skip connections to all layers as the input and intermediate dimensions are the same. For the second issue of the training degeneracy problem, it happens because the inputs lie in a low-dimensional subspace of the intermediate layer when applying skip connections to layers with width expansion. We apply the Hadamard transform to the second problem as it can not be solved by skip connections alone. We will rigorously define the second problem in the paper as you suggested.
> >
> > In addition, we want to mention that the zero-padding skip connection is one of the causes of the training degeneracy. Padding additional dimensions even with non-zero values still leaves the inputs in the low-dimensional subspace. This leads to the fact that: during the training, the weights connected to those additional dimensions are non-zero but strongly correlated. Obviously, checking whether zero weights are persisted during training is not enough, and thus measuring the forward and backward weight correlations is a more rigorous way.

---

> > > ### Author Response · Authors · 2021-11-19
> > > **Response to reviewer PQDc (Part 3)**
> > >
> > > >  (Quality) The paper concludes with results I consider to be negative. I thought this was interesting, as some of the changes made in this paper seem trivial and I did not expect them to harm performance. However, there was no proper ablation study (Figure 3 helped, but wasn't sufficient), so it remains unclear what element exactly caused the small degradation in performance. For example, I wonder if the single-convolution-skip architecture is responsible, but there is no way to disentangle the results from the entire zero-initialized-hadamard network.
> > >
> > > We would like to argue that our results are not negative. According to Table 1 in the paper, ZerO Init with ResNet-50 (AugSkip) is better than Xavier Init with standard ResNet-50 on ImageNet. Moreover, ZerO is better than Kaiming and Xavier in almost every setting under Resnet (AugSkip). We believe our work is also the first deterministic initialization for deep neural networks that achieves SOTA performance over various tasks.
> > >
> > > We agree that more ablation studies are needed to examine the reason for the slight degradation of Resnet (AugSkip). As shown in Appendix D.1, we followed your advice about more ablation studies on ResNet (Augskip). We evaluate the effects of our modifications to the original ResNet, including 1) adding additional skip connections and 2) applying the Hadamard transform. In Table 3 of Appendix D.1, the configuration with additional skips without Hadamard transform represents the “single-convolution-skip architecture” you are asking for. The results suggest that the slight accuracy degradation comes from the Hadamard transform instead of the additional skip connections (which means “single-convolution-skip architecture” is not responsible for the degradation). This also motivates us to find alternatives to break the training degeneracy while preserving the performance in the future.

---

> > ### Comment · Reviewer_PQDc · 2021-11-25
> > **Contribution 4**
> >
> > > We have conducted 5 more repeated experiments and updated the results in the revised paper. According to the latest result over 10 repeated runs, ZerO still consistently achieves smaller standard deviations than random methods.
> >
> > The insignificance of the presented result isn't just the result of using insufficient seeds, it's about what you report.
> >
> > When reporting the mean, you also report the standard deviation. This way, I can compare the differences in the mean with the reported standard deviation, to get a sense of how significant the result is (Ideally you would have p-values, but I was not asking or expecting that. The standard deviation is considered sufficient for there tasks). When you just report the difference in standard deviation- I have nothing to compare it with. Under the null hypothesis (Your new network did not affect the standard deviation at all), there is still a 50% chance that you would get a lower standard deviation, which would be true no matter how many seeds you used.
> >
> > You can't say that "ZerO still consistently achieves smaller standard deviations" in 10 repeated runs because standard deviation is, by definition, calculated for a series of experiments. This isn't a correct claim.

---

> > > ### Author Response · Authors · 2021-11-27
> > > **More reproducibility results are provided**
> > >
> > > Thanks for your clarification. We agree with you that we need to present reproducibility results in a more rigorous manner. To this end, we treat the empirical standard deviation $x$ of the test accuracy, measured over $m$ different trials of training as a random variable. We measure the variability of $x$ (mean and standard deviation) over $n$ different independent experiments. Note that $mn$ is the total number of independent training runs.
> > >
> > > We claim ZerO to be better in reproducibility if we have a lower mean for $x$ with comparable standard deviation compared to other schemes.
> > >
> > > For ZerO, we use the deterministic initialization and all randomness is due to different choices of mini-batches. We choose $m=5$ and $n=3$. The results are shown as follows:
> > >
> > > | Dataset | Model      | Initialization                            |Empirical standard deviation $x$ (mean $\pm$ std) | P-Value |
> > > |------------------|---------------------|----------------------------------------------------|:----:|:----:|
> > > | CIFAR-10         | ResNet-18 (AugSkip) | **ZerO Init**                                      | **$0.079 \pm 0.025$**       |         |
> > > |                  | ResNet-18           | Kaiming Init                                       | $0.123 \pm 0.019$           | $0.015$ |
> > > |                  | ResNet-18 (AugSkip) | Kaiming Init                                       | $0.144 \pm 0.026$           | $0.013$ |
> > > |                  | ResNet-18           | Xavier Init                                        | $0.135 \pm 0.018$           | $0.009$ |
> > > |                  | ResNet-18 (AugSkip) | Xavier Init                                        | $0.173 \pm 0.037$           | $0.008$ |
> > > | ImageNet         | ResNet-50 (AugSkip) | **ZerO Init**                                      | **$0.041 \pm 0.027$**       |         |
> > > |                  | ResNet-50           | Kaiming Init                                       | $0.069 \pm 0.032$           | $0.180$ |
> > > |                  | ResNet-50 (AugSkip) | Kaiming Init                                       | $0.086 \pm 0.041$           | $0.093$ |
> > > |                  | ResNet-50           | Xavier Init                                        | $0.103 \pm 0.053$           | $0.078$ |
> > > |                  | ResNet-50 (AugSkip) | Xavier Init                                        | $0.081 \pm 0.030$           | $0.075$
> > >
> > > According to the results, ZerO consistently achieves a smaller mean of empirical standard deviation $x$ compared to the other random methods, suggesting ZerO achieves better training reproducibility.
> > >
> > > **Update**: we compute p-values using t-test under two-sample equal variance assumption. For each random initialization, the null hypothesis is that its empirical standard deviation $x$ is equal to the $x$ under ZerO initialization, i.e., the initialization has the same reproducibility as ZerO. The p-value is computed as the probability of obtaining a test statistic greater than or equal to the observed value under the null hypothesis. We present the p-values for each random initialization in the table above. The results indicate that most null hypotheses are rejected with a p-value less than 0.1.

---

> > > > ### Comment · Reviewer_PQDc · 2021-11-28
> > > > **More reproducibility results are provided- Comment**
> > > >
> > > > Thank you for the response.

---

### Author Response · Authors · 2021-11-23
**General response to all reviewers and ACs**

We thank all reviewers for their thoughtful reviews. Here we provide a general response to clarify some common questions.

### Novelty & Contributions

We believe ZerO is the first successful deterministic initialization for training neural networks. Compared to random initializations, ZerO does not require a pre-defined distribution generating initial random weights. Thus, there is no need to assume the shape of the weight distribution or control the initial weight variance to ensure stable signal propagations.

Our main contributions include:
1. We discuss the difficulties when initializing weights to zero and propose solutions to address these difficulties.
2. We design ZerO that initializes ResNet with only zeros and ones deterministically. ZerO achieves state-of-the-art performance over various datasets.
3. We discover the advantages of ZerO from different perspectives. The results suggest that ZerO enjoys several unique benefits which are inaccessible for random initializations.

### Advantages
ZerO is a versatile initialization that has multiple benefits given its unique properties. With its “zero almost everywhere” property, ZerO shows advantages to convergence speed, training without batch normalization, and generating sparse networks through pruning. With its deterministic property, ZerO achieves better training reproducibility.

As some reviewers point out the comparison between Fixup and ZerO, we note that ZerO is not solely designed for normalization-free training. Instead, normalization-free training is one of its benefits. Compared to the random initializations (including Kaiming, Xavier, and Fixup), the advantages of generating sparse networks and training reproducibility are unique to ZerO due to its distinctive properties. We believe there will be more benefits of ZerO to be discovered in the future.

### Additional Experiments

During the rebuttal period, we conducted additional experiments given reviewers’ comments. The results are presented in Appendix C and D of the revised paper, which include:
1. ZerO for network pruning: we show that ZerO helps generate sparse networks with fewer parameters while preserving accuracy compared to random initializations.
2. Ablation study of ResNet (AugSkip): we evaluate and disentangle the effects of our modification to the original ResNet.
3. Analysis of learning rate warmup: we analyze why learning rate warmup is more important to ZerO than random methods. This is because the initial learning rate becomes the key factor of the gradient stability when the initial weight variance is inaccessible.
4. Additional repeated experiments: we repeated the whole experiment 5 more times (10 times in total) to ensure the statistical significance of our results regarding model stability. We updated the paper with the latest results.

### Summary

Overall, ZerO is a novel deterministic initialization outside the family of random initializations. Given the promising results and significant benefits of ZerO, we believe our work will guide future studies to replace conventional random initializations with deterministic methods for specific applications and scenarios.

---

> ### Author Response · Authors · 2021-11-23
> **Network Pruning**
>
> We discover a new benefit of ZerO for network pruning, and here we discuss its experiments and results. More detailed explanations of the experiments can be found in Appendix C of the paper.
>
> As we mentioned in Section 2.4, the final network trained with ZerO may have fewer useful parameters because most weights are extremely close to zero (refer to Figure 3 left). This motivates us to discover whether ZerO helps to generate sparse networks through pruning.
>
> To verify our hypothesis, we prune the trained networks initialized with different methods, including ZerO, Kaiming, and Xavier initializations. We then evaluate the performance of the pruned networks on the test set.
>
> Figure 6 in the paper shows the test accuracy of the pruned networks given different initializations and sparsity levels. Here we present a part of the results.
>
> **Pruned fully-connected network on MNIST**
>
> | Percent of Weights Remaining  |   30%   |   10%   |   5%  | 1% |
> | :---                          |   :----:| :----:  | :----:| :----:|
> | ZerO Init              |98.11%    |97.77%    |94.86%    |38.53%    |
> | Xavier Init            |97.45%    |81.01%    |61.95%    |20.58%    |
> | Kaiming Init           |95.79%    |75.92%    |54.51%    |20.29%    |
>
>
> **Pruned ResNet-18 on CIFAR-10**
>
> | Percent of Weights Remaining  |   80%   |   70%   |   60%  | 50% |
> | :---                          |   :----:| :----:  | :----:| :----:|
> | ZerO Init              |91.73%    |85.57%    |54.63%    |19.81%    |
> | Xavier Init            |89.41%    |55.3%    |30.35%    |16.05%    |
> | Kaiming Init           |87.21%    |61.71%    |30.48%    |10.51%    |
>
> As shown in the results, compared to Kaiming and Xavier initializations, the networks trained with ZerO can be pruned more aggressively while preserving the test accuracy. As the state-of-the-art pruning methods adopt standard random initializations by default, ZerO would be a powerful replacement that improves the pruning performance.

---

### Decision · Program_Chairs · 2022-01-20

**Decision:**

Reject

**Comment:**

This paper suggests an architecture with a deterministic initialization which has only 0/1 values.
The reviewers were mostly (marginally) negative, mainly because of the low novelty and significance of this work.

Specifically, the main novelty issues were:
1) Improving convergence speed and removing BatchNorm: was already done, in a quite similar manner, and it achieves better or similar results. (Fixup , ReZero: https://arxiv.org/abs/1901.09321, https://arxiv.org/abs/2003.04887, and few others as well)
2) Initializing a network with deterministic initialization: was also done (ConstNet, https://arxiv.org/abs/2007.01038). I think the main difference from the previous work is the additional Hadamard connections, which help break the symmetry. However, it is unclear what is the benefit of this modification, as the previous work could train without it (albeit on CIFAR).

Specifically, the main significance issues were:
1) Reducing standard deviation: The authors' response confirmed there is no statistically significant benefit (p=~ 0.1) for variance reduction when comparing with Kaiming initialization for ImageNet.
2) General network performance: The results do not seem better than the baseline (Xavier init is not a proper baseline in a network with ReLUs).
3) Sparsity claims: The network appears to be losing accuracy even with 20% sparsity, which isn't even useful for efficiency. For comparison, the lottery ticket hypothesis showed you can get to 90% sparsity and get better results. So, this is a nice observation, but not a major contribution.

Therefore, I recommend the authors to better distinguish themselves from previous works (What are the changes? Why are these important?), and improve their empirical results so they highlight the usefulness of the suggested method (e.g., improve the SOTA in some benchmark).